# Assessment of Microalgae Oil as a Carbon-Neutral Transport Fuel: Engine Performance, Energy Balance Changes, and Exhaust Gas Emissions

**Mantas Felneris [1,\*], Laurencas Raslavičius [1], Saugirdas Pukalskas [2] and Alfredas Rimkus [2]**

1   Department of Transport Engineering, Faculty of Mechanical Engineering and Design, Kaunas University of Technology, LT-51424 Kaunas, Lithuania; laurencas.raslavicius@ktu.lt
2   Department of Automobiles Engineering, Vilnius Gediminas Technical University, 03224 Vilnius, Lithuania; saugirdas.pukalskas@vilniustech.lt (S.P.); alfredas.rimkus@vilniustech.lt (A.R.)
\*   Correspondence: mantas.felneris@ktu.lt

**Abstract:** Notwithstanding the substantial progress acheved since 2010 in the attempts to realize the potential of microalgae biofuels in the transportation sector, the prospects for commercial production of $CO_2$-neutral biofuels are more challenging today than they were in 2010. Pure *P. moriformis* microalgae oil was subjected to unmodified engine performance testing as a less investigated type of fuel. Conventional diesel was used as a reference fuel to compare and to contrast the energy balances of an engine as well as to juxtapose performance and emission indicators for both unary fuels. According to the methodology applied, the variation of *BSFC* rates, *BTE*, smoke opacity, $NO_x$, HC, $CO_2$, $O_2$, and exhaust gas temperature on three different loads were established during compression ignition (CI) engine operation at EGR Off, 25% EGR, 18% EGR and 9% EGR modes, respectively. Simulation model (AVL Boost/BURN) was employed to assess the in-cylinder process parameters (pressure, pressure rise, temperature, temperature rise, *ROHR*, and *MFB*). Furthermore, the first law energy balances for an engine running on each of the test fuels were built up to provide useful insights about the peculiarities of energy conversion. Not depending on EGR mode applied, the CI engine running on microalgae oil was responsible for slightly higher *BTE* values, drastically reduced smoke opacity, higher $CO_2$ values, and smaller $O_2$ concentration, marginally increased $NO_x$ levels and lower total energy losses (in %) if compared to the performance with diesel fuel.

**Keywords:** microalgae oil; energy balance; renewable fuel; engine performance

## 1. Introduction

Over the past 40 years, sustainable and cleaner transport has been a target for many countries starting from the mid-1980s when the academic foundations of the "predict and provide" approach have been questioned in a number of comparative analyses of cities and their transport systems [1]. More or less at the same time, the modern biodiesel industry was established in Europe. Since then, biochemical platforms within biorefineries underwent a long way of development from the conversion of the first generation feedstock (that nowadays is described as unsustainable) to advanced ones, using biological and catalytic hydrocarbon pathways. As an alternative fuel, microalgae oils can offer almost all advantages of biodiesel: better lubrication properties of moving engine parts, high accumulated energy per unit volume, less stringent requirements for storing these fuels, performance indicators similar to or better than that of diesel fuel, and a significant reduction in atmospheric pollution. Even if the use of microalgae oils in internal combustion engines is not as well studied as biodiesel [2–5], the emission characteristics of these two fuels are very similar: an absence of organic sulfur compounds prevents the $SO_2$ release from the combustion process, sharp reduction in smoke, lower levels of carbon monoxide and a slight increase in nitrogen oxides. The main advantage of a microalgae oil-powered

vehicle over other alternative fuels is that such a car can be relatively called 'CO$_2$ neutral' (see Table 1). Such a definition cannot be applied to first-generation biofuels (Table 1) since they contain a certain amount of methanol derived from petroleum products, which is used in standardized technological processes [6]. The feature of CO$_2$ neutrality is characteristic to a very limited variety of alternative fuels. These are mainly synthetic fuels made from raw materials of biological origin by recycling carbon dioxide and combining it chemically with a substance of high chemical bond energy created from renewable energy [7].

**Table 1.** Life cycle GHG emissions for diesel fuel and biofuels from different oilseed crops [8,9].

| Oilseed Crops/Fossil Fuel | Soybean | Rapeseed | Microalgae Oil | Diesel Fuel |
|---|---|---|---|---|
| Life cycle GHG emissions | 49 | 37 | −183 | 93 |
| Demand for water use | High | High | Average | N/A |
| Demand for fertilizer use | Low-average | Average | Low | N/A |
| Demand for pesticide use | Average | Average | Low | N/A |
| Energy consumption | Low-average | Low-average | High | N/A |
| TRL (according to EU policy) | TRL 9—Competitive manufacturing in the case of key enabling technologies. Land use competition between foods and biofuels | TRL 9—Competitive manufacturing in the case of key enabling technologies. Land use competition between foods and biofuels | TRL 7—System prototype demonstration in an operational environment. There is potential for huge production volumes | TRL 9 |

Remarks: CO$_2$ amount (kg) emitted after the production of one MJ of energy (including cultivation, harvesting, processing, and combustion in an engine). For diesel fuel, GHG emissions were distributed in the following order: 22.11%—processing, transportation, production, 77.89%—fuel use stage.

If talking about the roadmaps for motorized transport in different countries from the broader concept of sustainability, the last decade has shown two trends: (i) the successful efforts of getting ahead in the race for market dominance for electric vehicles, that have reached the point where sales to 'early enthusiasts' started to give way to sales to a 'early majority' [10], and (ii) putting more and more emphasis on making diesel engines more powerful and efficient [11]. However, the fact that moving further and further out to the technological boundary is unacceptable for consumers, because each incremental step toward sustainability exploration and fuel economy is more costly and will adversely influence the air quality due to the increased mileage accumulated by new and more efficient vehicles [12,13]. It was estimated by EPA and NHTSA [12,13] that, in the case of USA, the pursuit of fuel economy standards would increase the cost of a new vehicle by more than €2000 and this difference in price would be passed by default on to the consumer who, most probably, will not be able to afford a new car. Doubtlessly, it brings some uncertainty and corrects the strategies of the original equipment manufacturers (OEMs) that currently are under the influence of a 'new realism'. The global trends prompt that advanced biofuels, especially made from algal biomass that do not compete with arable land and food could partially solve these problems by avoiding the development of OEMs products (vehicles) to become one-sided and too expensive for the customer without parallel development of bio-based carbon-neutral fuels as untapped opportunity to reach the goals of sustainability [14]. As described in [8], this approach would retain the use of low-cost internal combustion engines and liquid fuel systems which have high power densities and low embedded manufacturing and material extraction energies. These vehicles have considerable potential for further efficiency improvement [8]. These suggestions are illustrated by the fact that the global algae biofuel market (comprised of biodiesel, jet fuel, bioethanol, green diesel, etc.) for transport and aerospace applications was valued at approx. €4.1B in 2017 and is expected to generate a revenue of €8.7B by the end of 2024 with the compound annual growth rate 8.6% between this period [15,16]. In 2017, EU countries were among the leading players in the worldwide algae biofuel market with the leading country—Germany—owing to the presence of preponderant automobile

OEMs [15]. Moreover, in 2017 Finnish' ExxonMobil (formerly Mobil Oil Oy Ab) reported a breakthrough in research into advanced biofuels—the possibility to produce an algal biomass with obviously larger fat content which may become part of the energy mix for many counties in the distant future and will let them to cut $CO_2$ emissions significantly as well [17]. The introduction of microalgae oil brings a potentially carbon-free resource into the mix, in the form of renewable fuel for CI engines. In view of this, this present paper is mainly devoted to a comparison of energy balances, EGR effects, properties of combustion, operational performance, and emission characteristics of a CI engine fueled with microalgae oil *P. moriformis*, diesel fuel and their 30/70 and 70/30 blends.

## 2. Materials and Methods

Pure microalgae oil (MAO100) was subjected to unmodified engine performance testing as a less investigated type of fuel (Table 2). Conventional diesel fuel (D100) not containing a required 5% biodiesel additive was used as the reference fuel (Table 2) to compare and to contrast energy balances as well as performance and emission indicators for both unary fuels. In addition, to better evaluate the concept of diesel fuel's replacement level, two binary blends of 30/70 (D30/MAO70) and 70/30 (D70/MAO30) by volume also were a target of research in this study.

**Table 2.** Properties of unary fuels.

| Parameter | Diesel Fuel | Microalgae Oil |
|---|---|---|
| Fatty acid composition | – | Saturated: Lauric 12:0, Myristic 14:0, Palmitic 16:0, Stearic 18:0. Unsaturated: Palmitoleic 16:1, Oleic 18:1, Linoleic 18:2, Linolenic 18:3 |
| Density (at 15 °C), kg/m³ | 834.5 | 915.8 |
| LHV, MJ/kg | 42.80 | 36.91 |
| HHV, MJ/kg | 44.80 | 39.62 |
| Cetane number | 52.4 | 53.2 |

The Cetane number (CN) of microalgae oil was evaluated by using Equation (1) as reported in Ref [18]:

$$Cetane\ number\ (CN) = 46.3 + \left( \frac{5458}{\sum \left[ \frac{560 * A_i}{Mw_i} \right]} \right) - \left( 0.225 * \sum \left[ \frac{254 * D * A_i}{Mw_i} \right] \right), \quad (1)$$

where $A_i$ is the percentage composition of each fatty acid in the microalgae oil; $D$ is the number of double bonds present in each unsaturated fatty acid (see Table 2); $M_{wi}$ is the molecular weight of each fatty acid or its ester component. Parameters like molecular weight and molecular structure as well as methodological recommendations for CN, iodine value and saponification number calculations were used directly from Ref [18].

The engine test stand KI-5543 (Figure 1) is equipped with a weight dynamometer which is able to handle CI with a maximum brake load of 440 Nm. and a speed range of $n$ = 1600–3000 rpm. The engine control is realized by test bed automation, data acquisition is realized by an indication system using standard devices. The engine testing was performed under steady-state conditions, when the engine oil temperature reached the threshold value of 90 °C as well as stabilization of the exhaust gas temperature and the cooling water temperature was achieved. In each test, as described in [19], a fixed load was applied during 600 s, which included the stabilization period followed by data measurement and acquisition at steady state. Depending on the brake mean effective pressure (load) applied (*BMEP* = 0.4 MPa, 0.6 MPa, and 0.8 MPa), the stabilization period was 300–400 s; therefore, the time interval of data acquisition in a single test at a particular

load corresponded to 10.000–13.300 four-stroke engine cycles. More detailed description of the specific test procedures and methodologies applied is presented in Ref [20] that was published by co-authors in 2019. Table 3 shows the main engine characteristics. Each step of full-scale testing was repeatedly conducted for both cases of engine operation: EGR Off and EGR On. The uncertainties of the measurements are presented in Table 4.

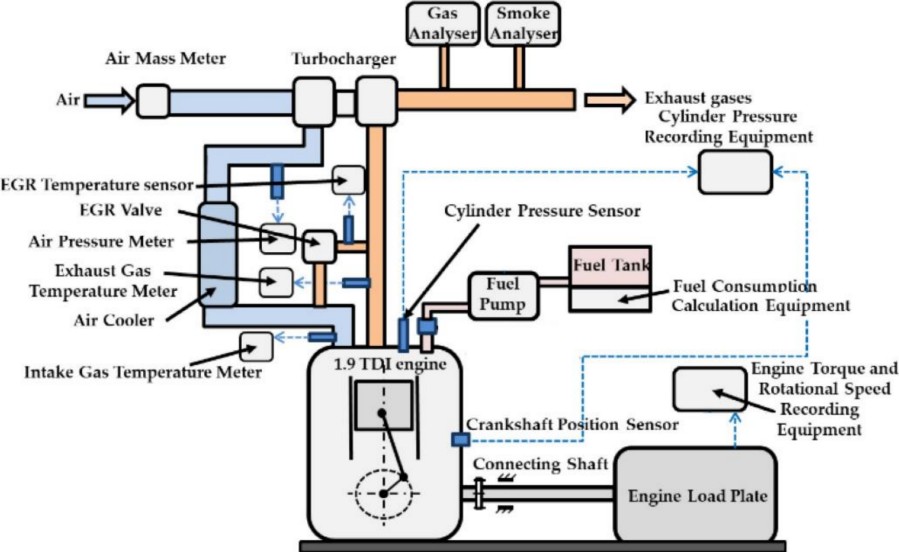

**Figure 1.** Engine test bench.

**Table 3.** Diesel engine specifications.

| Parameter | Type or value |
| --- | --- |
| Model | 1Z |
| Total displacement | 1896 cm$^3$ |
| SAC-hole nozzles | $5 \times 0.19 \times 150°$ |
| Compression ratio | 19.5:1 |
| No. of cylinders | 4 |
| Firing order | 1–3–4–2 |
| No. of strokes | 4 |
| Type of injection | Direct |
| Bore/stroke | 79.5 mm/95.5 mm |
| Maximum power at 4000 rpm | 66 kW |

**Table 4.** Uncertainty of the measured parameters.

| Parameter | Uncertainty |
| --- | --- |
| Brake load (KI-5543) | ±1.23 Nm |
| $CO_2$ (AVL DiCom 4000) | 0.1% vol. |
| HC (AVL DiCom 4000) | 1 ppm |
| $NO_x$ (AVL DiCom 4000) | 1 ppm |
| $O_2$ (AVL DiCom 4000) | 0.01% vol. |
| Smoke absorption (AVL DiCom 465) | 0.01 m$^{-1}$ |
| Fuel delivery time (VWR 61161-326) | 0.001% |
| Test fuel weight (SK-5000) | ±0.5 g |
| Exhaust gas temperature (K-type thermocouple) | ±0.5% |
| In-cylinder pressure (AVL GH13P) | ±0.3% |

Integral to the test bench's engine management system (electronic control unit), EGR valve recirculated the metered quantities of exhaust gas mass to the engine intake system.

At different engine loads (*BMEP* = 0.4 MPa, 0.6 MPa, and 0.8 MPa), the EGR coefficient was $k_{EGR}$ = 0.25, 0.18 and 0.09, respectively. The calculation of the $k_{EGR}$ was performed according to Equation (2):

$$k_{EGR} = \frac{m_{EGR}}{m_{EGR} + m_{air\_EGR}}, \tag{2}$$

where $m_{EGR} = m_{air} - m_{air\_EGR}$ is the mass of recirculated gases introduced to engine cylinders, kg/h; $m_{air\_EGR}$—the mass of fresh intake air in the total intake mixture, when EGR is On, kg/h; $m_{air}$—the mass of fresh intake air at EGR Off mode, kg/h.

The in-cylinder pressure was recorded by an AVL GH13P piezo sensor, which was integrated in the preheating plug and sent signals to a data acquisition system through the AVL DiTEST DPM 800 amplifier. Numerical analysis of the in-cylinder processes was accomplished for the engine numerical model in AVL-BOOST environment. The captured traces of in-cylinder pressure coupled with fuel and air consumption measurement data, physico-chemical parameters of test fuels, and engine specification data were further employed for the assessment of the rate of heat release (*ROHR*), in-cylinder temperature, temperature rise, pressure rise and mass fuel burnt (*MFB*). The steps to develop a simulation model is presented in [20] that was earlier published by the co-authors.

The first-law energy balance for an engine which is based on the energy conservation principle, provides useful information about the conversion of chemical energy in the fuel into heat and then—into mechanical energy. The implied formulation has the structure [21]:

$$Q_f = Q_B + Q_{cool.} + Q_{ex.} + Q_{rad.} + Q_{i.c.,} \tag{3}$$

where $Q_f = m_f \cdot LHV$—chemical energy stored in the fuel (input energy), MJ; $Q_B$—brake power energy, MJ; $Q_{cool.}$—energy transferred to the cooling medium, MJ; $Q_{rad.}$—radiation from the engine's external surfaces, MJ; $Q_{i.c.}$—enthalpy flux of the exhaust gases, MJ; $m_f$—fuel mass, kg; *LHV*—lower heating value of a fuel, MJ/kg.

From Equations (4)–(6) it is easy to deduce the output energies that are relevant to this study:

$$Q_B = Q_f \cdot BTE, \tag{4}$$

$$Q_{ex.} = m_{ex} \cdot c_{p\_ex} \cdot T_{ex} - \left( m_{air} \cdot c_{p_{air}} \cdot T_{air} + m_f \cdot c_f \cdot T_f \right), \tag{5}$$

$$Q_{cool.} \approx Q_f - Q_B - Q_{ex.} - Q_{rad.,} \tag{6}$$

where *BTE*—brake thermal efficiency; $c_p$—specific heat capacity, MJ/kg K; *T*—temperature, K.

Radiation heat losses from a diesel engine's external surface are very difficult to measure directly. Hence, our study relies on the assumption that $Q_{rad.}$ comprises 4% of the overall output energy. Energy losses from incomplete combustion due to the lack of combustion air are usually very small in diesel engines and thus can be neglected ($Q_{i.\,c.} = 0$). Energy balance for each constituent percentage in the four groups was determined by the following equation:

$$q_n = Q_n \cdot 100 / Q_f. \tag{7}$$

## 3. Results

### 3.1. In-Cylinder Processes

Commonly in CI engines, the four stages in the combustion process are as follows: ignition delay period, premixed combustion phase (all fuel has been injected, and the pressure increases rapidly), mixing controlled combustion phase (the fuel is burned and produces power), and late combustion phase (the pressure is going down). In this section, the impact of microalgae oil on engine combustion characteristics is discussed in terms of in-cylinder pressure, pressure rise (*PR*), *ROHR*, *MFB*, and in-cylinder temperature and temperature rise rate.

Figure 2a illustrates the in-cylinder pressure and pressure rise diagrams of an engine at 2000 rpm and average load by fueling with microalgae oil and diesel fuel in two

selected operating conditions: EGR Off and 18% EGR. It could be seen in this diagram that comparing with diesel operation, the peak in-cylinder pressure became a little higher with microalgae oil fueling. The same figure demonstrates the behavior of the pressure curves obtained for 18% EGR for the selected load. Due to the reduction of intake pressure and *MFB* (Figure 2b), the in-cylinder pressures for both unary fuels were reduced through the compression and power strokes of diesel operating cycle, with a lower maximum level [20]. Until the very end of the ignition delay period, a dip in pressure rise can be noticed (Figure 2a), because both D100 and MAO100 are evaporating and therefore consume energy instead of releasing it.

The rates of heat release curves for D100 and MAO100 are shown in Figure 2b, coupled with two derivative combustion parameters—$ROHR_{MAX}$ (the maximum value of heat flux) and $HR_{PREMIX}$ (the heat released during the premixed phase of combustion) [22]. This phase ranges between the start of combustion (at 3.5 degCA) and the CA corresponding to $ROHR_{MAX}$ (14.5 degCA for D100 and 16.0 degCA for MAO100) [22].

The early stage of the premixed combustion phase is associated with the rapid increase in cylinder pressure (3.5 ... 6.5 degCA) showing the highest peaks for diesel fuel not depending on EGR activation/deactivation. On the other hand, PR (Figure 2a), ROHR (Figure 2b) and temperature rise (Figure 2c) curves show the influence of 18% EGR on the marginal reduction (by ~0.5 degCA) in the ignition delay, especially for MAO100. The main combustion outlined a different development, with slightly higher levels of $ROHR_{MAX}$ and $HR_{PREMIX}$ for microalgae oil combusted under EGR Off mode while under the 18% EGR mode both unary fuels demonstrated similar values. Nevertheless, only a qualitative evaluation is possible, while these parameters had to be evaluated for a more detailed analysis [22]. At the crank angles corresponding to the particular volume of 10% (5.8 degCA for D100 and 6.5 degCA for MAO100), 50% (15.0 degCA for both D100 and MAO100) and 90% (37.0 degCA for D100 and 34.0 degCA for MAO100) of the *MFB*, the combustion duration (*MFB* 90–10) and its impact on exploitation parameters of an engine are assessed for the sufficient explanation of the experimental results that were extensively discussed in Sections 3.2 and 3.3.

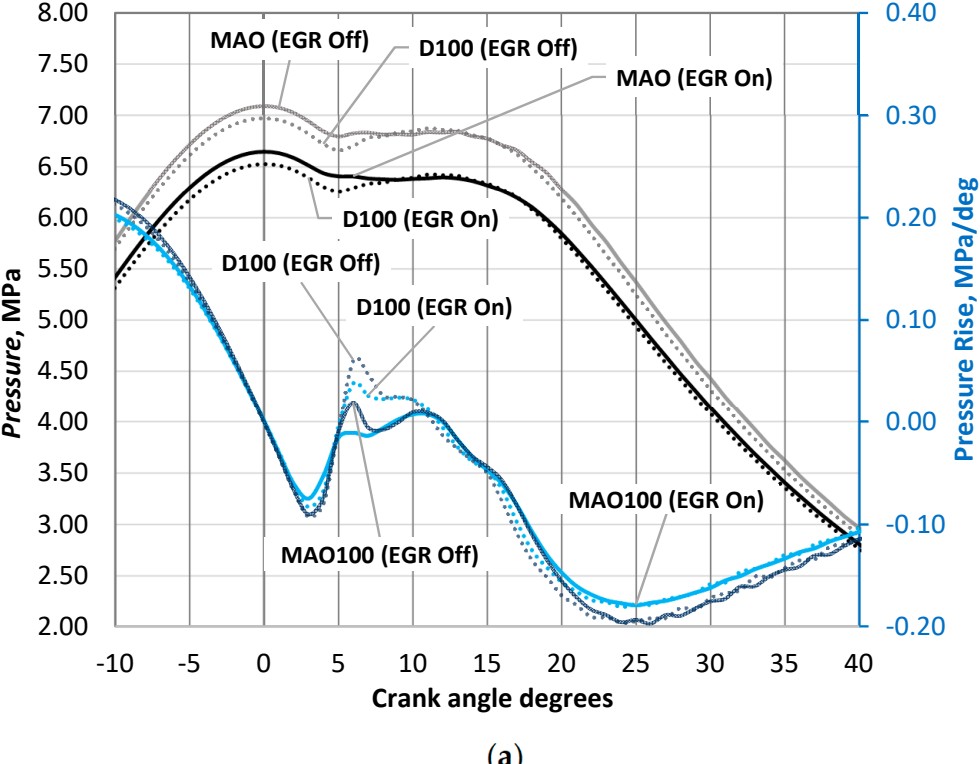

(a)

**Figure 2.** *Cont.*

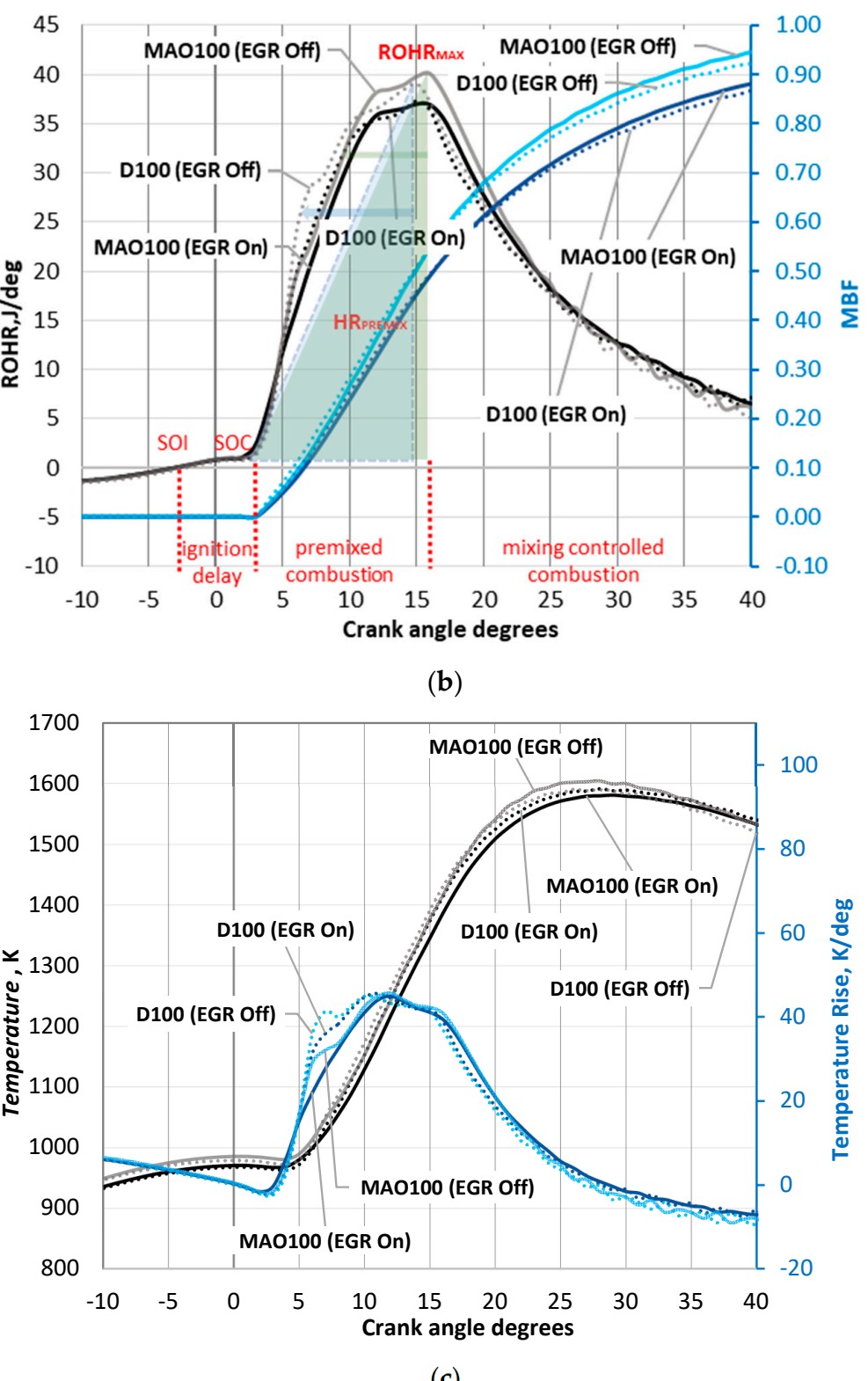

**Figure 2.** Variation of in-cylinder pressure and pressure rise (**a**), *ROHR* and *MFB* (**b**), in-cylinder temperature and temperature rise (**c**) as functions of crankshaft position (*n* = 2000 rpm, *BMEP* = 0.6 MPa).

As can be seen from the in-cylinder temperature distribution chart (Figure 2c), the maximum values of 1604 K at 28 degCA for MAO100 and 1591 K at 26 degCA for diesel fuel are achieved under simulation study, thus being the highest temperature values under the air running condition. At 18% EGR mode, diesel fuel demonstrates the highest peak

temperature value of 1592 K at 28 degCA while microalgae oil is responsible for 1581 K at 29 degCA. These characteristic points correspond to *MFB* = 0.84 (EGR Off) and *MFB* = 0.78 (18% EGR) for microalgae oil and *MFB* = 0.78 (EGR Off) and *MFB* = 0.76 (18% EGR) for diesel fuel, respectively. Temperature rise curves follow the pattern of *ROHR* and pressure rise simulation outcomes showing a simultaneous increase in their values at ~6.5 degCA and ~14.5–16.0 degCA not depending on the type of fuel used (Figure 2c). As a consequence, these trends coincide well with the depicted $ROHR_{MAX}$ and $HR_{PREMIX}$ values.

## 3.2. Engine Operational Performance

In the present subchapter, the engine performance characteristics were evaluated considering different fuels, different engine loads, and activation/deactivation of exhaust gas recirculation valve as input parameters, while *BSFC*, *BTE* and exhaust gas temperature were discussed as the output parameters.

### 3.2.1. Brake Specific Fuel Consumption (BSFC)

Figure 3a shows the variation of *BSFC* at 0.4 MPa, 0.6 MPa, and 0.8 MPa load conditions for different tested fuels and different EGR ratios. It can be seen that the maximum *BSFC* rates were achieved at 0.4 MPa load that corresponds to the low load mode. The inferior heating value of microalgae oil decreases the calorific value of the binary blends with the increase in the biofuel percentage ratio as the difference between LHV values for pure diesel fuel (44.80 MJ/kg) and MAO (39.62 MJ/kg) comprises 13.76%. These numbers suggest that practically it is essential to ensure that in 15.96% bigger amounts of fuel are delivered to each engine cylinder when microalgae oil is burnt.

The power output regulation of a CI engine is achieved by varying the fuel supply and, due to the nature of the combustion process, it always runs on lean and nonhomogeneous air-fuel mixtures. The larger amount of atmospheric air being induced into the engine cylinders leads to higher in-cylinder pressure and temperature rates (Figure 2a,c), higher *MFB* and *ROHR* (Figure 2b) that clearly indicates during the bench tests obtained higher *BSFC* rates for oxygenated types of fuel [20]. The lowest *BSFC* rates were observed for pure D100 (238–264 g/kWh) and for the blend with 30% biofuel concentration (+4.08 . . . +4.20%) while the biggest increase was observed for a higher concentration of microalgae oil in the range of 70% fuel blend (+9.47 . . . +10.50%) and for MAO100 in its pure form (+13.64 . . . +15.13%).

Throughout the present study, the EGR-rates of 0.25 (at 0.4 MPa), 0.18 (at 0.6 MPa), and 0.09 (at 0.8 MPa) are used as they give information about the effects on the combustion process in a more direct way. During the EGR On mode, the amount of mass flow of fresh air in the intake piping is directly coupled to the amount of EGR that the engine can run at a particular load, because the limiting factor is the air/fuel ratio in the cylinder. The EGR Off and EGR On modes were compared at a number of load points (Figure 3a) and it is shown that the amount of EGR has the potential to increase the engine *BSFC*. The maximum increase was 18.70% (MAO100) for operation with $k_{EGR}$ = 0.09 and average load, the minimum—was 2.10% (D100) for operation with $k_{EGR}$ = 0.09 and high load (Figure 2a). Figure 2c exhibits the variation of exhaust gas temperature with and without EGR mass fraction for diesel fuel and microalgae oil. It is found that with $k_{EGR}$ = 0.09, the exhaust gas temperature slightly reduces that lead to a pressure drop during premixed combustion and mixed control combustion phases (Figure 2a) followed by a considerably lower pressure rise rate, which decreases the rate of combustion of the fuel. This effect is predetermined by the lower intake temperature due to a higher cooling capacity in the long-route system, as the EGR passes both EGR-cooler and the intercooler [23].

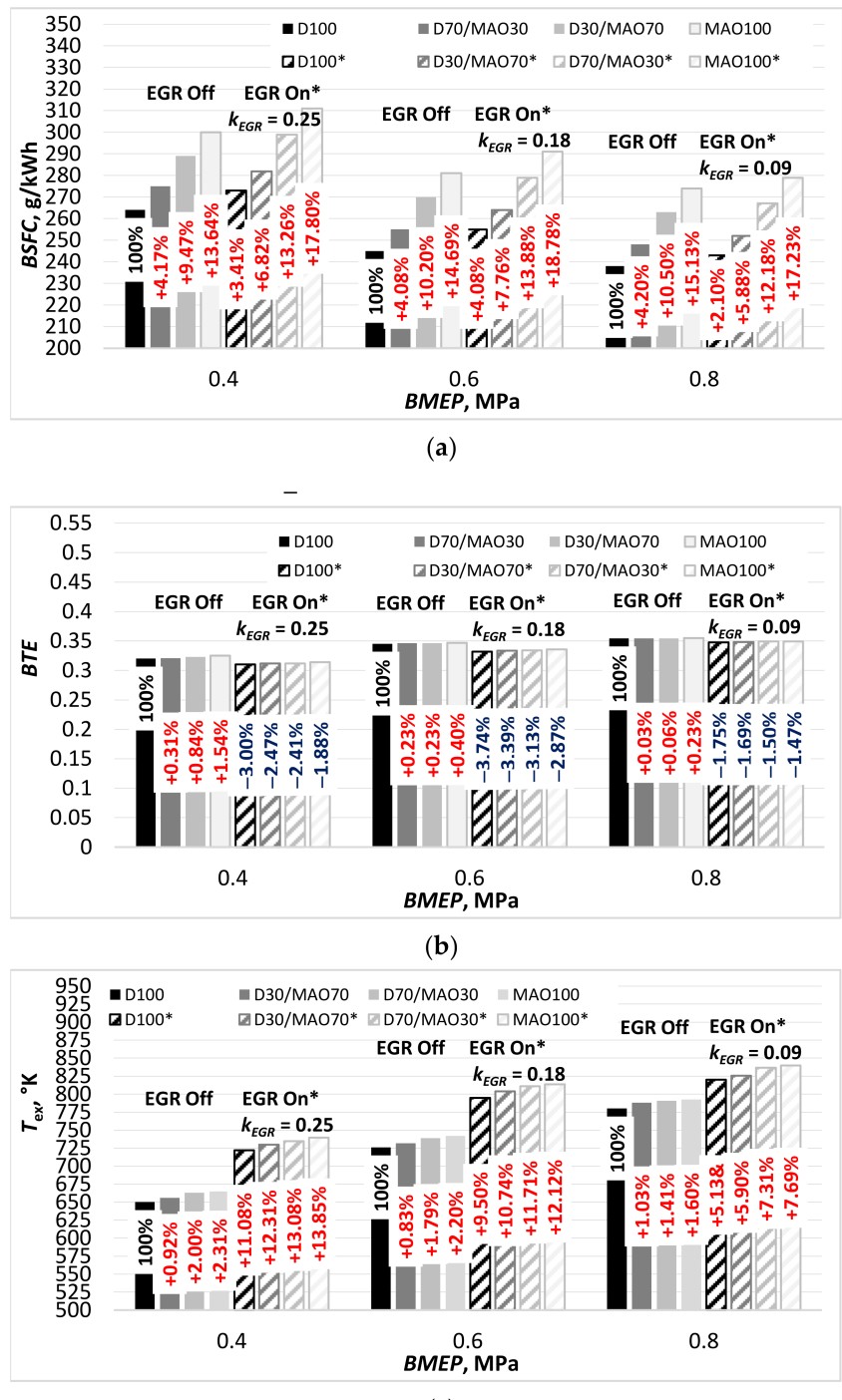

**Figure 3.** Variation of energy indicators of a diesel engine running on the investigated types of fuel and using manual deactivation/activation of EGR valve: *BSFC* rate against *BMEP* (**a**); *BTE* value against *BMEP* (**b**); $T_{ex}$ value against *BMEP* (**c**).

### 3.2.2. Brake Thermal Efficiency (BTE)

Figure 3b shows the trends of *BTE* vs. engine load and EGR rate for the investigated types of fuel. The results indicate that when the test engine was running at EGR Off mode and was powered by oxygenated fuels, the brake thermal efficiency slightly increased compared to D100, and vice versa—on average a 0.03–1.54% decline in *BTE* values was observed for selected EGR rates not depending on the type of fuel used. This trend is inversely proportional to the *BSFC'* character (see Figure 3a,b). It can be seen that the use

of microalgae oil leads to the highest brake thermal efficiency (0.355) among all tested fuels at high load condition, followed by a slight decrease in *BTE* values at 0.6 MPa (0.347) and 0.4 MPa (0.325) loads. At high load condition, the fuel injection pressure reaches its peak such that the viscosity effects are negligible, and the presence of 10.5% of fuel-borne oxygen in the chemical composition of MAO100 [20] enables a more complete combustion and hence higher *BTE*. At 0.4 MPa engine loading, *BTE* values were found to be 0.320, 0.321, 0.323, and 0.325 with D100, D70/MAO30, D30/MAO70 and MAO100, respectively, which were increased by approximately 7.94–8.31% at average load, followed by additional increase by 2.58–2.81% at high load condition. The combustion of microalgae oil resulted in higher peak pressure compared to D100 (Figure 2a) during the ignition delay phase that led to an enhanced spray atomization into small droplets, vaporization and mixing with air. Generally, in the described operating conditions, the mass fuel burning of 90% with fossil diesel fueling was 3.5 degCA postponed in comparison with that with MAO100 fueling (Figure 2b), since more fuel was burnt during the late combustion phase when MAO100 was adopted. This indicates a shorter combustion duration for microalgae oil. Coupled with the higher *ROHR* (Figure 2b) obtained during the mixing controlled combustion phase, the aforementioned in-cylinder processes enhanced the engine's brake thermal efficiency when unary and binary oxygenated fuels were used.

A detailed insight into the response of the tested fuels to EGR is plotted in Figure 3b. It is evident that the exhaust gas has little influence on the combustion characteristics at low EGR ratios. This is because the recirculation of exhaust gases, however, displaces much of the necessary air for high-temperature chemical reactions within the cylinders, hence leading to a decrease in *BTE* [24]. As depicted in Figure 2a, at $k_{EGR}$ = 0.18, the peak pressures are apparently decreased by 6.1–8.5% in comparison with the operation with no EGR. Another negative aspect is that the specific heats of $CO_2$ and $H_2O$ are slightly higher than that of the air [25], which reduces the overall combustion temperature (Figure 2c). It means that the proportion of molecules composed of three atoms increases when more products of combustion are recycled in the combustion system. For the high specific heats of the triatomic molecules the recirculated gas has, more heat would be absorbed and the total heat released will decrease [25,26]. The general conclusion to be drawn from the above analysis is that the microalgae oil exhibits the lesser vulnerability to EGR if compared to D100. As a result, combustion of pure microalgae oil tolerates higher EGR rates than diesel fuel without significant deterioration of *MFB* [26,27].

### 3.2.3. Exhaust Gas Temperature

The variation of exhaust gas temperature with respect to the load and EGR rates to compare D100 and microalgae oil is shown in Figure 3c. The temperature at the exhaust manifold increases with the increase in load when more heat is generated due to the burning of more fuel. Burning of larger injection volumes to meet the high load condition results in excess heat loss through exhaust gases.

This is a consequence of the longer injection duration that shortens the premixed combustion phase, thus prolonging a mixing controlled combustion phase. This in turn makes a direct impact on the maximum temperature to be reached inside the combustion chamber. For all fuel cases examined at EGR Off mode, the differences in exhaust gas temperature are more pronounced at low load mode, showing an increase by 2.31% during the combustion MAO100. This difference tends gradually to decrease to 2.2% for average and to 1.6% for high load mode, thus indicating that the combustion process, nevertheless of the fuel used, remains stable and comparable to diesel fuel. The increase in the exhaust gas temperature with increasing of the blending ratio of microalgae oil may be also associated with the fact that diesel fuel burns faster during the much shorter phase of premixed combustion (approx. 3.0–14.5 degCA) in comparison to MAO100 which loses the biggest part of its mass during the much longer phase of mixing controlled combustion. The shift of thermochemical reaction's intensity towards the later phases, including after burning is

a consequence of poorer atomization as well as slower and longer combustion of MAO100 that creates slightly higher exhaust temperatures.

The cylinder pressure vs. crank angle for the CI combustion at average load with EGR fraction of 18% is shown in Figure 2a. Moreover, as provided in Figure 2b, the in-cylinder temperature and *MFB* reduces not depending on the type of fuel used. This reflects an effective utilization of heat energy because the energy of the same fuel amount is thus used to heat up a working fluid with the larger heat capacity than it would without EGR. In other words, the decrease in oxygen content within the cylinder charge reduces the combustion rate leading to retarded combustion and thus to lower peak cylinder pressure values (Figure 3a). Putting more fuel in the cylinder at higher engine load, the endothermic dissociation reactions of the EGR components such as water vapor will also cool down the combustion chamber. However, the decrease in overall cylinder temperature through the use of recirculated exhaust gases is not primarily reflected in Figure 3c. The trend of higher exhaust gas temperature rates for EGR On mode is explained by a factor that was already described above—the shift of combustion reaction's intensity towards the later phases. This trend is noticeably pronounced in Figure 2c where the curves of heat release vs. degCA for the late combustion phase expose higher values compared to EGR Off mode. The temperature released during the late combustion phase actually is comparable to the temperature entering the exhaust gas valve. That's why variations in $T_{ex}$ for numerous fuels at different loads and EGR On mode show higher values in comparison to EGR Off mode.

### 3.3. Emission Characteristics

The emission data was analyzed and presented graphically for smoke opacity, $NO_x$-specific emissions, HC, $CO_2$, and $O_2$ (Figure 4). The uncertainties of the measurements are presented in Table 4.

### 3.3.1. Smoke Opacity

The effect of microalgae oil on smoke opacity is presented in Figure 4a. The results obtained show that all of the 30/70 and 70/30 (vol./vol.) blends, and especially microalgae oil, notably reduced soot concentrations with respect to conventional diesel, a finding that comes in agreement with previous research [20,28]. The decreasing trend of soot concentration in the exhaust gases of microalgae oil-containing fuels are mostly associated with better carbon-oxygen balance leading, to an improvement in the combustion reaction and better promotion of soot oxidation, especially at the end of the cycle. This conclusion is also confirmed by the findings presented in Figure 4d, whereas the decreasing trend of $CO_2$ gases for microalgae oil indicates the better completeness of the combustion reaction when biofuels are used.

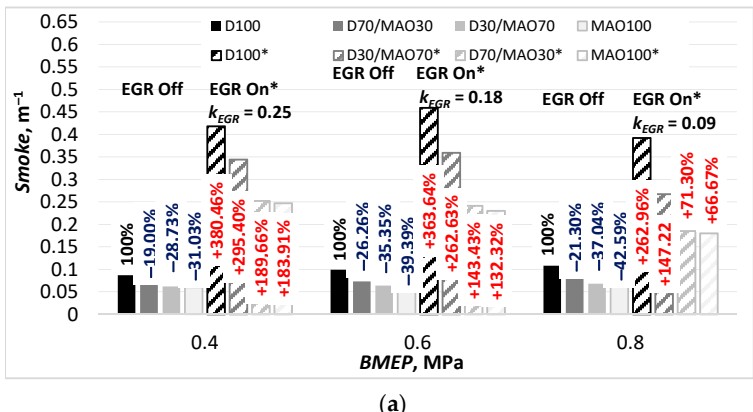

(a)

**Figure 4.** *Cont.*

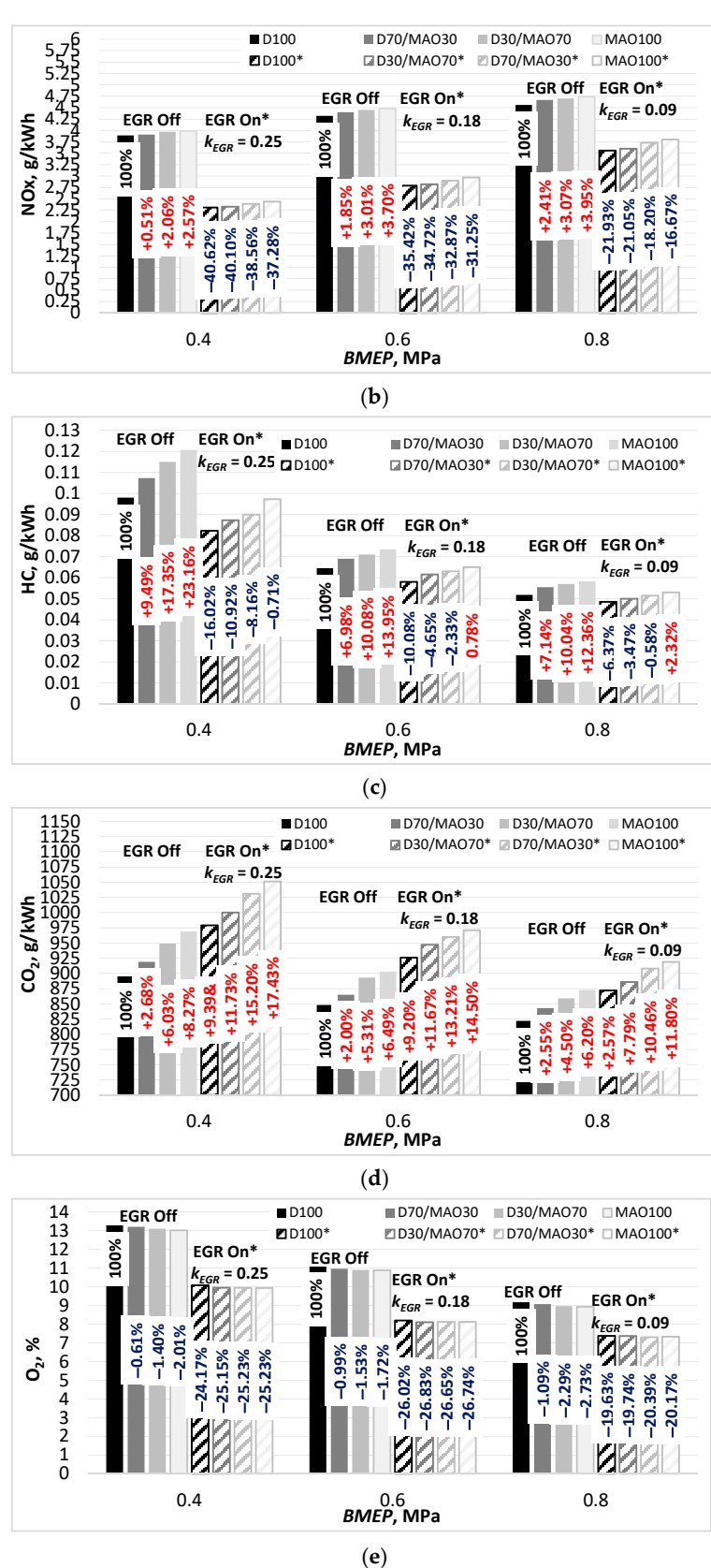

**Figure 4.** Variation of ecological indicators of a diesel engine running on investigated types of fuel and using manual deactivation/activation of EGR valve: smoke opacity rates against *BMEP* (**a**); NO$_x$ values against *BMEP* (**b**); HC values against *BMEP* (**c**); CO$_2$ values against *BMEP* (**d**); O$_2$ values against *BMEP* (**e**).

As described in [29], it is already known that during fuel combustion in a diesel engine, the effect of EGR causes the increase of the smoke opacity concentration. This is associated with an extended combustion duration (rate of combustion) that directly affects the amount of oxidized soot and, finally, the net soot release. It is seen from Figure 2b that the *MFB* and *ROHR* rates for the tested fuels at 18% EGR were markedly lower during the whole cycle of combustion if compared to the obtained ones at EGR Off mode that gives an indication about the extended process of combustion. In practice, this means that at the time the exhaust valve opens, soot combustion is still in progress. The opening of the exhaust valve causes the soot oxidation process to stop prior reaching its final stage and the products of incomplete combustion to enter the exhaust manifold. At *BMEP* = 0.4 MPa and 25% EGR, we observed an almost fourfold increase in smoke concentration for diesel fuel and a twofold—for microalgae oil. Depending on the biofuel volume ratio in the blend, the binary fuels demonstrated in 295.4% and 189.7% higher pollution levels for the same load. The decrease in smoke opacity with the increase in microalgae oil volume ratio in the blend is associated with a lower number of carbon atoms and presence of in-built oxygen in the chemical composition of a particular biofuel. Diesel fuel and microalgae oil were responsible for the highest smoke opacity concentrations at 25% EGR and low load ($0.418$ m$^{-1}$ vs. $0.247$ m$^{-1}$) as well as 18% EGR and medium load ($0.459$ m$^{-1}$ vs. $0.230$ m$^{-1}$) modes, respectively leading to a slight reduction at *BMEP* = 0.8 MPa due to reduction in EGR rate.

### 3.3.2. NO$_x$

Nitric oxide (NO$_x$) and particulate matter are the main pollutants of diesel engines [30]. The diesel baseline measurements consisted of three operating conditions, equally distributed over the operating range (Figure 4b). The highest NO$_x$ emissions occur at $n$ = const. = 2000 rpm and *BMEP* = 0.8 MPa when temperature and residence time were greatest. The NO$_x$ emission at *BMEP* = 0.8 MPa was 4.56 g/kWh, a value slightly higher than the baseline emissions for diesel operation at *BMEP* = 0.6 MPa (4.32 g/kWh). As seen in Figure 4b, NO$_x$ values for microalgae oil and its blends with diesel fuel show a marginally increasing trend at all loads. At *BMEP* = 0.4, MPa NO$_x$ was produced at very lean conditions compared to stoichiometric conditions. It can be suggested that the presence of excess air at lean conditions, coupled with comparatively lower in-cylinder pressure and temperature effects during the premixed combustion phase, enabled N$_2$ to react with hydrocarbon groups to form NO$_x$ via the Prompt NO$_x$ mechanism [31]. At average and high loads of an engine, the in-cylinder temperatures for MAO100 were slightly above 1600 K (see Figure 2c) that led to shorter ignition delay and lower *ROHR*. It can be hypothesized that for the higher loads Thermal NO$_x$ mechanism was much more dominant than the Prompt NO$_x$ [31], however, no sharp increase in NO$_x$ emissions due to the presence of inbuilt oxygen in biofuels was further observed.

The drop in temperature (see Figure 2c) due to the displacement of the fresh air with recirculated exhaust gases led to an obvious reduction in NO$_x$ emissions (Figure 4b) what is consistent with the current understanding of NO$_x$ formation and the observations of other researchers [32]. As described in [33], there are three reduction mechanisms postulated in this regard:

- Chemical mechanism that was already described in the above chapter states that increased dissociation of the more complex triatomic molecules reduces the heat released.
- Thermal mechanism—the increased heat capacity of the triatomic molecules in the recirculated gases results in lower flame temperatures.
- Dilution mechanism—the potentially increased mixing time and longer burn dilution caused by the dilution effect of the EGR results in lower flame temperatures.

The NO$_x$ emissions at half load (0.6 MPa) and 2000 rpm decreased by 35.42% for D100 and 31.25% for microalgae oil at 18% EGR. At small load, the NO$_x$ emissions showed a similar behavior to half load. They decreased even more drastically by 40.62% for diesel fuel and 37.28% for MAO100. Since more air is needed to burn the higher amount of fuel at

*BMEP* = 0.8 MPa, the maximum attainable $NO_x$ reduction at 9% EGR rate at this load is lower compared to that at 0.4 and 0.6 Mpa.

### 3.3.3. HC

HC formation for the particular type of diesel engine is not problematic as can be seen in Figure 4c. Usually, HC generation most likely occurs during incomplete combustion when there is a substantial lack of oxygen in the combustion chamber. However, the deficiency of oxygen was not observed as fuel combustion was always running with excess air and the fuel is burned almost completely. Most likely, the main source of hydrocarbon emissions is caused by the injector sac volume—a small fuel portion that is left at the end of the injection [24]. These droplets injected at a very low velocity remain in the immediate vicinity of the nozzle tip and thus undergo slow evaporation and combustion [23,34]. This leads to a slow mixing with air and thus some fuel can escape the combustion because locally over-lean mixtures will not auto-ignite or support a propagating flame. This fact is especially distinctive to MAO100 and its binary blends with diesel fuel since these fuels have higher viscosity (what means higher resistance to flow) in comparison to D100. The sac causes injectors to deliver uncalibrated excess fuel when the engine is operated under closed throttle as well as under idle or low load conditions [35].

Regarding HC-specific emissions, the highest values were found for all tested fuels at low load mode as well as pure microalgae oil was the most distinguished among other fuels in terms of the highest emission levels. The maximum decrease recorded at *BMEP* = 0.6 MPa was 39.1% for MAO100, 38.3% for D30/MAO70, 35.7% for D70/MAO30 and 34.2 for D100. By switching the engine to run on high load mode, the further decrease in HC levels was 20.8% for MAO100 and 19.6–19.7% for D30/MAO70, D70/MAO30 and D100, respectively.

Decreased HC emissions were observed with the use of 25% EGR and *BMEP* = 0.4 MPa load, 18% EGR and *BMEP* = 0.6 MPa load, and 9% EGR and *BMEP* = 0.8 MPa load. It has been already discussed that a major source of hydrocarbons emissions in diesel engine is the undermixed fuel, quantity of which primarily depends on the injected volume. Figure 2b clearly indicates that the *MFB* values for D100 and MAO100 were significantly lower for 18% EGR and *BMEP* = 0.6 MPa load than those obtained for EGR Off and *BMEP* = 0.6 MPa load. The same trend also indicates that the *MFB* rates for MAO100 are slightly higher in comparison to that demonstrated by D100 that corresponds very well with the experimental findings depicted in Figure 4c.

### 3.3.4. $CO_2$

The trend presented by $CO_2$ (Figure 4d) is, in fact, a reflection that carbon dioxide emissions can be used as a first indication for fuel consumption (considering also the fuel's HHV and LHV) [30]. Nevertheless, biofuels have a lesser amount of carbon atoms in their chemical composition if compared to the traditional chemical formula for petroleum diesel that is $C_{12}H_{26}$, they usually expose higher *BSFC* and $CO_2$ rates. This can be explained through the analysis of the findings given below. Figure 2a presents the percentage change of mass-based fuel consumption of the engine when using microalgae oil, compared to market diesel. An increase of mass-based fuel consumption is observed with all biofuels, ranging from 4.08 to 15.13%. Theoretically, it is explained by the fact that the inferior heating value of microalgae oil decreases the calorific value of the blend with increasing air-fuel ratio. As referred to in [36], the stoichiometric ratio (*SR*) for a diesel oil of HHV = 45.70 MJ/kg is approximately 14.42. Zhu and Venderbosch [36] present a correlation between *SR* and HHV for different types of fuel represented by:

$$SR = 0.31 \text{HHV} \tag{8}$$

In this equation, the ultimate element analysis was substituted by the caloric value test. By using this mathematical expression, we obtained a 11.57% difference in *SR* values for two different fuels: $SR_{D100}$ = 13.89 (HHV = 44.80 MJ/kg) and $SR_{MAO100}$ = 12.28

(HHV = 39.62 MJ/kg). We need to denote that the difference between LHV values for pure diesel fuel and MAO100 reaches 13.76%. These numbers suggest that theoretically it is essential to ensure that in ~16% bigger amount of fuel is delivered to the engine cylinders followed by an additional 0.35 kg amount of oxygen per 1.16 kg of microalgae oil burnt. The traditional chemical formula for petroleum diesel is $C_{12}H_{26}$ and its density is 0.835 t/m$^3$ [20], which results in approx. 300.16 g of $CO_2$ by 100 g of diesel, according to $1C_{12}H_{26} + 18.5O_2 + 69.56N_2 \rightarrow 12CO_2 + 13H_2O + 69.56N_2$ [37]. From the stoichiometric combustion reaction of microalgae oil with air and taking into account its density of 0.916 t/m$^3$, the $CO_2$ emission rate would be approx. 294.67 g of $CO_2$ per 100 g of biofuel burnt. If adding the already discussed 15.96% to 294.67 g, we will obtain 341.70 g of $CO_2$ emitted by the microalgae oil-powered engine to maintain the same power. These facts coincide very well with the AVL simulation outcomes for D100 and MAO100, which show the increasing in-cylinder pressure and temperature rates (Figure 2a,c) as well as higher *MFB* and *ROHR* (Figure 2b) when microalgae oil is used. The introduction of the exhaust gases in the inlet charge (some of the $O_2$ is replaced by $CO_2$) resulted in a further increase in engine $CO_2$ emissions (Figure 4d).

### 3.3.5. Oxygen Content

The concentration of certain exhaust gas constituents alone without giving a clear description about their interaction with other pollutants is not indicative of the contribution of harmful compounds to the overall atmospheric pollution prevention because of the wide variations in exhaust gas chamber geometry, different engine types and different operating conditions [38]. The determination of the oxygen level as the exhaust gas constituent is necessary since all emission measurements and juxtaposition must be done for a given reference level of $O_2$. The typical oxygen level for a standard diesel engine running at various loads and $n$ = 1500–3000 rpm exhaust composition is 6.0% to 15.5%-vol. [39,40]. Figure 4e shows $O_2$ emissions emitted by all test fuels. Among all test fuels, diesel fuel recorded the marginally highest $O_2$ emissions at $n$ = 2000 rpm. Meanwhile, the emissions by MAO100 decreased on average by 2.15% compared to diesel fuel. The $O_2$ emission trends for both unary fuels are contextualized with the experimental findings presented in Figures 3b and 4a,d and lead to a conclusion that microalgae oil has a very similar or even better completeness of the combustion reaction when compared to diesel fuel. During complete combustion, C and $H_2$ combine with $O_2$ to produce $CO_2$ (↑ for MAO, ↓ for D100—see Figure 4d) and $H_2O$. During incomplete combustion, part of the carbon is not completely oxidized producing smoke (↓ for MAO, ↑ for D100—see Figure 4a). Finally, this conclusion is confirmed by the findings presented in Figure 3b, showing marginally higher *BTE* values for a diesel engine running on unary and binary biofuels. The use of EGR results in lower exhaust oxygen emissions which were very similar for both fossil and renewable fuels and fluctuated in a range of 24.17–25.23% at 25% EGR, 26.02–26.74% at 18% EGR and 19.63–20.17% at 8% EGR, respectively. Depending on the load applied, MAO demonstrated in 1.06%, 0.72%, and 0.54% lower $O_2$ emissions, thus explaining the lesser drop in *BTE* values and in 5.3–8.05% higher $CO_2$ emissions if compared to conventional diesel.

### 3.4. Energy Balance Assessment

This section focuses on a comparative energy balance (in MJ and in %) for a 4-cylinder turbocharged direct injection CI engine operating on D100, MAO100, and their D70/MAO30 and D30/MAO70 blends as well as using manual deactivation/activation of EGR valve. Steady-state tests were run to experimentally determine how the input energy in the form of fuel was appropriated throughout the compression-ignition engine. The transfer of energy was measured for the engine's effective power ($Q_B$), losses to the engine coolant ($Q_{cool}$) and exhaust ($Q_{ex}$), and losses by radiation ($Q_{rad}$). The main goal for studying how microalgae oil, diesel fuel, and their binary blends transferred their energy throughout the same system was to help draw conclusions on the performance, variation of in-cylinder processes, effectiveness of combustion, lubrication and other characteristics of fossil diesel

compared to the microalgae oil. In each of the charts a-h (Figure 5), the variation of the energy balances at $n$ = 2000 min$^{-1}$ for low, medium, and high loads is presented. We assume the amount of input energy needed to produce a certain brake power as constant for all test fuels and EGR Off mode: 45.2 MJ for *BMEP* = 0.4 MPa, 67.9 MJ for *BMEP* = 0.6 MPa and 90.5 MJ for *BMEP* = 0.8 MPa. If convert to percentages, microalgae oil is responsible for the highest $Q_B$ values that surpass the same parameter for diesel fuel in 12.02% (low load), 7.85% (average load) and 8.68% (high load), respectively (Figure 5b,h). What is specific to the EGR Off mode is that diesel fuel has almost equally distributed cooling and exhaust energy losses for all loads, each of which comprises roughly 30% of the total energy output (Figure 5h). With the increase of biofuel ratio in the blend, up to pure microalgae oil, the diesel engine releases a lesser amount of cooling losses but a higher amount of exhaust loss (Figure 5b,d,f). This is in line with the above findings on *ROHR*, *MFB*, and in-cylinder temperature rates for both unary fuels (Figure 2b,c) clearly showing that microalgae oil is responsible for the longer duration of the combustion process and thus higher exhaust gas temperatures than D100. The binary blend MAO70/D30 stands out from the other test fuels in terms of an almost unchanged amount of cooling losses during engine operation at different loads that fluctuated at approx. 25–26% (Figure 5f). The rise of brake mean effective pressure, followed by the increase of combustion temperature, causes the overall increase of all constituents of the energy balance: $Q_B$, energy losses for cooling, $Q_{ex}$, and $Q_{rad}$ (Figure 5a,c,e,g). If consider a percentage part out of 100%, then $Q_{rad}$ = const. = 4.0% and does not depend on the load applied, while the $Q_{ex}$ values tend to decrease with the rise of *BMEP* (Figure 5b,d,f,h). The total output energies (EGR Off mode) of the unary fuels in an ascending order are 127.6 MJ and 143 MJ (*BMEP* = 0.4 MPa), 182.8 MJ and 197.2 MJ (*BMEP* = 0.6 MPa), and 233.3 MJ and 262.2 MJ (*BMEP* = 0.8 MPa) for MAO100 and diesel fuel, respectively (Figure 5a,g).

Activation of EGR valve predetermines slightly increased total output energy (Figure 5a,c,e,g) with a tendency of $Q_B$ values being highest for MAO100 and lowest for D100. This trend coincides well with the marginal reduction in *BTE* values at different EGR rates for diesel fuel and slightly higher brake thermal efficiency indications for unary and binary microalgae fuels (Figure 3b). Not depending on the type of fuel used, the total output energy increases in the following range: 2.32–3.45% for low, 0.71–4.46% for average and 1.17–3.39% for high load modes. During engine operation with a manually activated EGR valve, MAO100 was responsible for the biggest difference of 3.45% for low load (from 127.6 MJ to 132.0 MJ), while diesel fuel demonstrated the 4.46% increase for medium (from 197.2 MJ to 206.0 MJ) and 3.39% (from 253.6 MJ to 262.2 MJ) for high load modes (Figure 5a,c,e,g). The difference in percentages between $Q_B$ values obtained at EGR Off and EGR On modes (with emphasis on the latter) for the same engine load is explained by up to 21.12% bigger amount of energy being lost for engine cooling followed by up to 11.7% decrease in exhaust losses (Figure 5b,d,f,h).

The biggest differences between $Q_{cool}$ and exhaust $Q_{ex}$ values at EGR Off and EGR On modes are obtained at 25% EGR and 18% EGR, respectively (Figure 5b,d,f,h). This is in line with the results plotted in Figure 2a–c, where *ROHR* and *MFB* trajectories for MAO100 and D100 have lower temperatures at a given pressure when 18% EGR dilution is added.

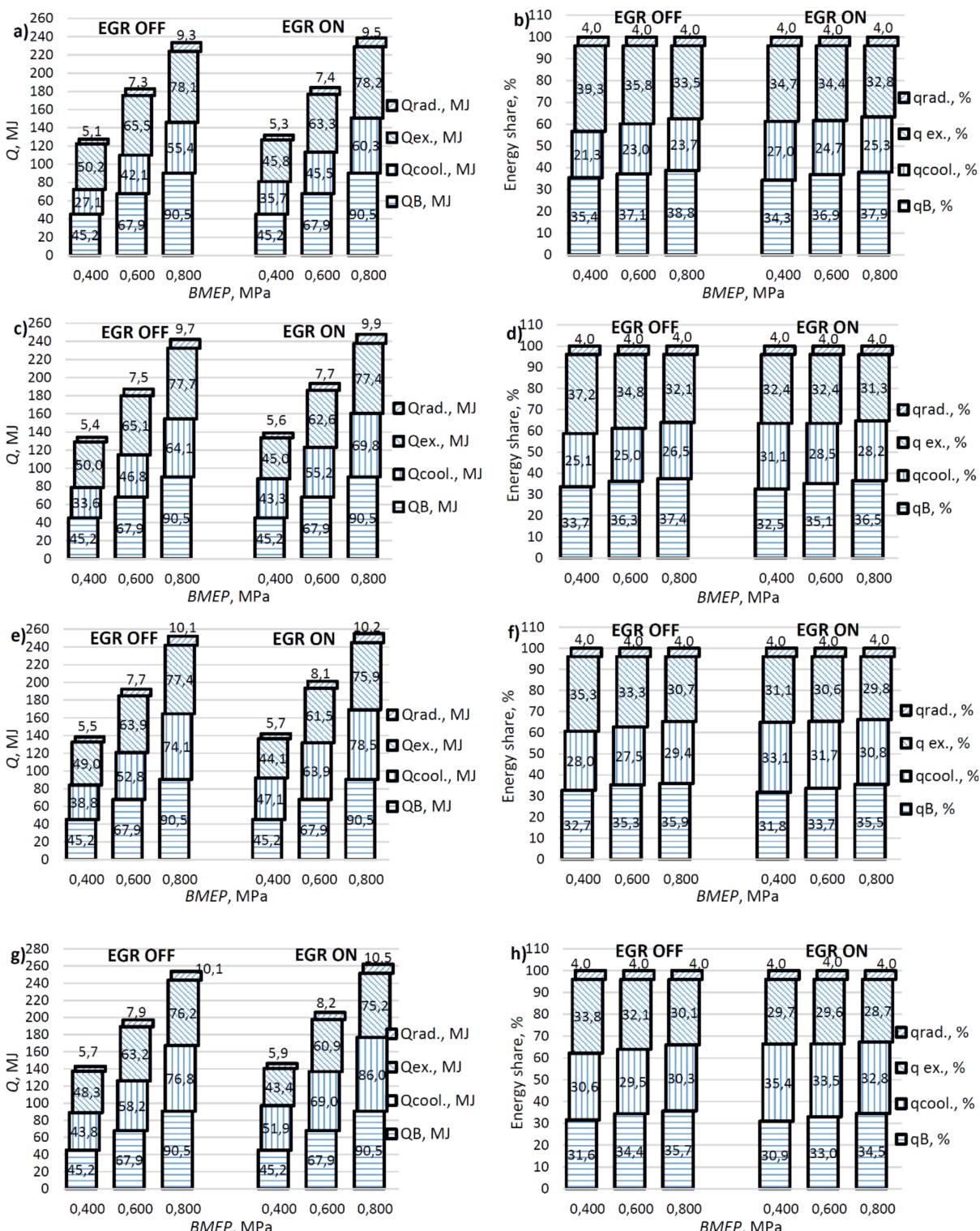

**Figure 5.** Variation of the total input and output energies (in MJ and in %) for three different loads of a diesel engine running on the investigated types of fuel and using manual deactivation/activation of EGR valve: MAO100 (**a,b**), D30/MAO70 (**c,d**), D70/MAO30 (**e,f**) and D100 (**g,h**).

## 4. Conclusions

The following conclusions were drawn from this investigation that used a four-cylinder, direct-injected, turbocharged, CI engine running with diesel fuel, microalgae oil, and their 30/70 and 70/30 (vol./vol.) blends:

(1) Due to higher density and lower energy content, microalgae oil distinguishes for a longer injection duration compared to diesel fuel that predetermines higher values of heat release rate during the mixing controlled combustion phase what, at the same time, positively influences the better burn out of the solid particles and the overall decrease in smoke opacity.

(2) The higher values of in-cylinder pressure and in-cylinder temperature facilitate the ignition (heats the unburned portion of the charge) and shorten the delay before its ignition. In the case of microalgae oil as a product of biological origin having high oxygen content, this helps to avoid the sharp increase in $NO_x$ emissions.

(3) Microalgae oil produces a much more intensive reaction zone during the mixing-controlled combustion phase, which, as a consequence, leads to a more complete combustion process and better fuel efficiency (mass burnt per energy produced).

(4) Results revealed that the exhaust gas recirculation (EGR) system reduces the in-cylinder temperature that can be explained by the following facts:

- Lower heat release rate, in-cylinder temperature, and air-fuel ratio acted in favour of $NO_x$ reduction as well as prevented the formation of this harmful compound;
- In the case of prolonged combustion, more heat from the combustion gases is transferred to and conducted through the cylinder walls that increases the amount of heat energy delivered through the cooling system in the form of heat loss. This lead to an increase in fuel consumption.

(5) The fuel energy rate transferred to the engine's effective power as well as lost energy rate were calculated for each fuel and then compared to each other:

- At EGR-Off mode, microalgae oil demonstrated on average in $-10.8\%$ (BMEP = 0.4 MPa), $-7.3\%$ (BMEP = 0.6 MPa) and $-11.0\%$ (BMEP = 0.8 MPa) lower energy losses (in MJ) if compared to diesel fuel;
- At 18% EGR mode, MAO100 demonstrated on average in $-3.3\%$, $-4.3\%$ and $-3.3\%$ lower energy losses than D100 for the same loads, respectively.

(6) The study has found that microalgae oil and its 30/70 and 70/30 blends with conventional diesel are more or less equally sensitive to key engine parameters compared with pure diesel fuel, and potentially can be adopted to the CI engine of a passenger car, type 1Z, as well as to the entire family of industrial diesel engines [41].

**Author Contributions:** Conceptualization, M.F., L.R., S.P. and A.R.; methodology, M.F., L.R., S.P. and A.R.; software, A.R.; validation, A.R.; formal analysis, M.F. and L.R.; investigation, M.F., L.R., S.P. and A.R.; resources, S.P. and L.R.; writing—original draft preparation, M.F. and L.R.; writing—review and editing, M.F.; visualization, A.R. and L.R. All authors have read and agreed to the published version of the manuscript.

**Funding:** This research received no external funding.

**Institutional Review Board Statement:** Not applicable.

**Informed Consent Statement:** Not applicable.

**Acknowledgments:** The analysis of combustion process indicators was performed using the engine simulation tool AVL BOOST, acquired by signing a Cooperation Agreement between AVL Advanced Simulation Technologies and the Faculty of Transport Engineering of Vilnius Gediminas Technical University (VILNIUS TECH).

**Conflicts of Interest:** The authors declare no conflict of interest.

## Nomenclature

**Latin symbols**

| | |
|---|---|
| $A_i$ | Percentage composition of each fatty acid in the microalgae oil |
| $c_p$ | Specific heat capacity (MJ/kg K) |
| $D$ | Number of double bonds present in each unsaturated fatty acid |
| $HR_{PREMIX}$ | Heat release during the premixed phase of combustion |
| $k_{EGR}$ | EGR coefficient |
| $m_{air}$ | Mass of fresh intake air at EGR Off mode (kg/h) |
| $m_{air\_EGR}$ | Mass of fresh intake air in the total intake mixture when EGR is On (kg/h) |
| $m_{EGR}$ | Mass of recirculated gases introduced to engine cylinders (kg/h) |
| $M_{wi}$ | Molecular weight of each fatty acid or its ester component |
| $m_f$ | Fuel mass (kg) |
| $n$ | Engine speed (rpm) |
| $Q_B$ | Brake power energy (MJ) |
| $Q_{cool}$ | Energy transferred to the cooling medium (MJ) |
| $Q_{i.c.}$ | Enthalpy flux of the exhaust gases (MJ) |
| $Q_f$ | Chemical energy stored in the fuel (input energy) (MJ) |
| $Q_{rad.}$ | Radiation from the engine's external surfaces (MJ) |
| $ROHR_{MAX}$ | Maximum value of heat flux (J/degCA) |
| $SOC$ | Start of combustion (degCA) |
| $SOI$ | Start of injection (degCA) |
| $T$ | Temperature (K) |
| $T_{ex}$ | Exhaust gas temperature (K) |

**Abbreviations**

| | |
|---|---|
| *BMEP*: | Brake mean effective pressure (MPa) |
| *BSFC*: | Brake specific fuel consumption (g/kWh) |
| *BTE*: | Brake thermal efficiency |
| *CA*: | Crankshaft rotation angle |
| CI: | Compression ignition |
| CN: | Cetane number |
| $CO_2$: | Carbon dioxide (g/kWh) |
| D100: | Pure diesel fuel |
| D70/MAO30 | Binary blend comprised of microalgae oil (30%) and diesel fuel (70%) |
| D30/MAO70 | Binary blend comprised of microalgae oil (70%) and diesel fuel (30%) |
| degCA: | Crankshaft rotation angle degrees |
| EGR | Exhaust gas recirculation |
| GHG: | Greenhouse gases |
| HC: | Hydrocarbons (g/kWh) |
| HHV: | Higher heating value (MJ/kg) |
| HRR: | Heat of release rate (J/degCA) |
| LHV: | Lower heating value (MJ/kg) |
| MAO100: | Pure microalgae oil |
| MFB: | Mass fuel burnt |
| $NO_x$: | Nitrogen oxides (g/kWh) |
| $O_2$: | Oxygen (%) |
| OEM: | Original equipment manufacturers |
| ROHR: | Rate of heat release (J/degCA) |
| $SO_2$: | Sulfur dioxide |
| SR: | Stoichiometric ratio |
| TRL: | Technology readiness level |

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
