# Peer review of "Assessment of Microalgae Oil as a Carbon-Neutral Transport Fuel: Engine Performance, Energy Balance Changes, and Exhaust Gas Emissions"

_sustainability, doi:10.3390/su13147878_

Round 1

Reviewer 1 Report

The authors tested pure P.moriformis microalgae oil like type of fuel.  To compare  the energy balances of engines, they used conventional diesel as a reference. The comparison was made for a four-cylinder, direct-injected, turbocharged, CI engine.

The authors made complex research and presented it in complicated form. I suggest to make the text a more clear for non-professional engineer reading. For example, many abbreviations are used and a list of abbreviations should be added.

I also miss brief discussion of advantages and disadvantages of the application of pure P.moriformis microalgae oil like type of fuel

Author Response

Reviewer #1: The authors tested pure P.moriformis microalgae oil like type of fuel.  To compare the energy balances of engines, they used conventional diesel as a reference. The comparison was made for a four-cylinder, direct-injected, turbocharged, CI engine.

Corresponding author: I would like to thank to Reviewer#1 for the careful and detail review. Answers and some additional explanations for the mentioned queries are given below.

Reviewer #1: The authors made complex research and presented it in complicated form. I suggest to make the text a more clear for non-professional engineer reading. For example, many abbreviations are used and a list of abbreviations should be added.

Corresponding author: Necessary changes have been made. At the end of the manuscript we included the Abbreviations list to make the reading of the paper more comfortable (changes are marked in red font).

Reviewer #1: I also miss brief discussion of advantages and disadvantages of the application of pure P.moriformis microalgae oil like type of fuel.

Corresponding author: Necessary changes have been made – we included additional bullet point (#6) to the conclusions section:

The study has found that microalgae oil and its 30/70 and 70/30 blends with conventional diesel are more or less equally sensitive to key engine parameters compared with pure diesel fuel, and potentially can be adopted to the CI engine of a passenger car, type 1Z, as well as to the entire family of industrial diesel engines [41].

 The paper is now revised substantially according to the reviewers’ comments (highlighted in a PDF file). We took on board all comments and have included necessary information. Hence, all comments below have been directly or indirectly addressed.

Reviewer 2 Report

The manuscript was written well and the science content is error-free. Hence I recommend its publication.

Author Response

Reviewer #2: The manuscript was written well and the science content is error-free. Hence I recommend its publication.

Corresponding author: I would like to thank to Reviewer#2 for the positive review.

Reviewer 3 Report

The authors studied an interesting and important topic. The detailed discussion, together with detailed graph illustrations, makes it much easier to follow and understand. The work could  bring some better understanding for  the related research groups.  The manuscript is well-organized and well-written. I recommend the manuscript to be published at current form!

Author Response

Reviewer #3: The authors studied an interesting and important topic. The detailed discussion, together with detailed graph illustrations, makes it much easier to follow and understand. The work could bring some better understanding for the related research groups.  The manuscript is well-organized and well-written. I recommend the manuscript to be published at current form!

Corresponding author: I would like to thank to Reviewer#3 for the positive review.